# DigiHEALTH: Suite of Digital Solutions for Long-Term Healthy and Active Aging

**DOI:** 10.3390/ijerph20136200

**Published:** 2023-06-22

**Authors:** Cristina Martin, Isabel Amaya, Jordi Torres, Garazi Artola, Meritxell García, Teresa García-Navarro, Verónica De Ramos, Camilo Cortés, Jon Kerexeta, Maia Aguirre, Ariane Méndez, Luis Unzueta, Arantza Del Pozo, Nekane Larburu, Iván Macía

**Affiliations:** 1Fundación Vicomtech, Basque Research and Technology Alliance (BRTA), Mikeletegi 57, 20009 Donostia-San Sebastian, Spain; 2Faculty of Engineering, University of Deusto, Avda. Universidades, 24, 48007 Bilbao, Spain; 3Biodonostia Health Research Institute (Bioengineering Area), eHealth Group, 20014 Donostia-San Sebastián, Spain

**Keywords:** older adults, active aging, healthy lifestyle, digital solutions, voice assistant, face recognition, gait analysis, risk prediction, recommendation system, well-being assessment

## Abstract

The population in the world is aging dramatically, and therefore, the economic and social effort required to maintain the quality of life is being increased. Assistive technologies are progressively expanding and present great opportunities; however, given the sensitivity of health issues and the vulnerability of older adults, some considerations need to be considered. This paper presents DigiHEALTH, a suite of digital solutions for long-term healthy and active aging. It is the result of a fruitful trajectory of research in healthy aging where we have understood stakeholders’ needs, defined the main suite properties (that would allow scalability and interoperability with health services), and codesigned a set of digital solutions by applying a continuous reflexive cycle. At the current stage of development, the digital suite presents eight digital solutions to carry out the following: (a) minimize digital barriers for older adults (authentication system based on face recognition and digital voice assistant), (b) facilitate active and healthy living (well-being assessment module, recommendation system, and personalized nutritional system), and (c) mitigate specific impairments (heart failure decompensation, mobility assessment and correction, and orofacial gesture trainer). The suite is available online and it includes specific details in terms of technology readiness level and specific conditions for usage and acquisition. This live website will be continually updated and enriched with more digital solutions and further experiences of collaboration.

## 1. Introduction

The expectancy of life is increasing worldwide at the same time that medical science is improving and being able to provide a wider spectrum of health services [1]. Together with that, other professionals, such as nutritionists, physiotherapists, psychologists, etc., are also improving our quality of life and providing (together) a holistic understanding of care services that are centered on persons [2,3].

Nowadays, computers support human input, decision making, and provision of data. In today’s healthcare sector and medical profession, AI, optimization algorithms, big data, or language processing are used to derive inferences for monitoring long-run behavioral trends and detecting and measuring individual associated risks and opportunities based on data-driven estimations [4]. The healthcare sector highly depends on data and analytics to improve therapies, practices, service personalization, and adherence. The global (North America, Europe, Asia Pacific, Latin America, and Middle East and Africa) AI market in healthcare was approximately $1.4 billion in 2018 and is expected to reach approximately $17.8 billion by 2025, with a compound annual growth rate (CAGR) of 43.8% between 2019 and 2025 [5]. In North America, the AI market size in healthcare exceeded $1.15 billion in 2020 and is expected to witness over 44.2% CAGR from 2021 to 2027. However, in the European context, AI in the healthcare market size exceeded $700 million in 2020 and is forecasted to witness a growth rate of 43.9% from 2021 to 2027 [6].

However, some important challenges are related to encouraging technology utilization [7] and minimizing technological barriers [8] that older people are frequently facing [9]. Analyzing technological frontiers and leveraging the potential of data in healthcare services is a tremendous challenge [10]. In this respect, data privacy is especially sensitive when talking about health. User credentials and authentication processes should be facilitated [11] so that we ensure secure access to digital services and guarantee the correct use of personal data. This can be accompanied by introducing technology that is friendly to the final user and gets closer to the experience of human interaction. The introduction of language processing mechanisms, for example, may be very well received as part of a digital ecosystem for older adults [12,13,14].

The debate about the definition of ‘quality of life’ for older individuals is conducted among researchers from various disciplines and overlaps with explorations of the concepts such as successful aging, subjective well-being, life satisfaction, and happiness [15]. The concept of a full and meaningful life has been adopted by the World Health Organization [16] as well as several governments and organizations. It implies not only physical health but also psychological aspects, level of independence, social relationships, and environment. Experts recognize that this is facilitated by several aspects that include physical activity, nutritional aspects, sleep routines, etc. Some previous studies revealed that self-assessment tools that can (or not) include simple recommendations [7] can help tremendously to promote daily routines and healthy lifestyles [17,18].

Holistic patient-based approximations include assistive technologies that improve the care trajectory all along the patient journey, including phases or episodes with different professionals (doctors, physiotherapists, etc.). Specific digital services are designed to help professionals and improve the portfolio of services provided, reducing the overall costs and/or professional efforts. In a wide spectrum of impairments and diseases, very different and specific tools to predict heart failure decompensation, improve facial gestures, or rehabilitate specific movements [19,20] have shown to be very useful.

The World Health Organization’s digital health strategy emphasizes the potential for digital health to transform global health and improve the health of all people [21]. However, there are still challenges to overcome before the widespread benefits from data, AI, and digital health can be realized [22,23]. The democratic use of information requires the harmonization of technical [24] and governance standards [25]. Despite the increasing use of health-related information, gaps remain in the ability of systems to exchange and use information, defined as interoperability [26]. Data are collected in ways not universally recognized, with conceptually different values recorded under the same term. Although there has been steady progress toward the development of universal standards, the adoption of these standards by healthcare systems remains unclear [27]. Similarly, inconsistent regulatory and governance approaches hinder progress because legal requirements differ across data types, the purpose of use, and jurisdictions. Federal data spaces provide the tools and methods to provide more flexible and adaptable digital services [28]. Federal data spaces build on data sovereignty and refer to the authorized relationships among multiple actors that present specific boundaries of decision-making authority, rights, roles, and responsibilities around data processing [29].

All in all, and despite the huge amount of health data that is being produced [30] the predictive, preventive, and personalized medicine aspiration outlined by Healthcare 4.0 remains intangible [31]. This is due to the unavailability of the right health data at the right time to data consumers, i.e., the researchers, medical practitioners, machine learning algorithms, and policymakers [32]. In this scenario, several suites and marketplaces have been launched with the idea of connecting healthcare data consumers with the producers. The health industry has proposed several health data suites. ATOS [33] has launched a digital suite as a flexible health data platform that builds upon big data analysis tools and artificial intelligence to provide enhanced services for health management. The digital suite of Philips [34] offers a selection of tailor-made tools and resources for the innovation and cocreation of custom applications. It builds on interoperability between applications and devices and provides predictive analytics and decision-making algorithms as a result of machine learning algorithms. Other industries that work on the digitalization of health services have proposed marketplaces as a digital space to interchange specific APIs (application programming interface) with specific health or well-being purposes. Healthineers’ Digital Marketplace, the proposal of SIEMENS [35], provides an open and secure environment for a wide range of healthcare stakeholders to share, access, and analyze data. It is the entry point for connecting users with a portfolio of applications that connect solution partners and application-exploiting consumers. As a secure online health connector, MyChart, available by EPIC Health [36], provides a seamless application for self-management of health that includes health data storage and visual analytics, sharing and connecting possibilities, health agenda management options, etc. Navify Marketplace, launched as a beta version by ROCHE [37], enables easy access and acquisition of digital products and services from Roche and third parties. This life portfolio includes specific applications such as test readers, monitoring tools for specific analyzers, or systems for actively predicting reagents and consumable needs. An innovative marketplace proposal is provided by IQVIA [38], a centralized portal of developer tools for healthcare and the life sciences. The site includes three main tools: (a) IQVIA HealthCare Locator SDK can connect any mobile or web app to any healthcare database; (b) a platform to enable customization and integration of client instances of OCE (orchestration customer engagement) APPs; and (c) NLP API SUITE, healthcare natural language processing set of APIs that focus on improving the usability of unstructured text.

The development of marketplaces specifically designed for older individuals remains unclear, and no result has been found when looking for marketplaces or suites inspired by reflexive and shared knowledge-generating processes. This participatory approach is crucial for untapping the potential of health data and deep into the possibilities of sharing specific solutions for older adults. Therefore, this paper presents DigiHEALTH, which provides a suite of digital solutions that includes digital tools in three specific chapters that aim to facilitate (a) authentication processes that ensure data privacy; (b) long-term healthy and active aging; (c) improved health services that aid the caregiving for specific impairments. The presentation of this digital suite tries to approach a wide spectrum of professionals alleviating the well-known distance between research and innovation context and market requirements. This suite is the result of a long trajectory where health-related research has moved forward in close collaboration with professionals in digital technologies, health services, sociosanitary affairs, and especially, working hand in hand with older people.

## 2. Materials and Methods

This paper presents a suite of digital solutions that have been codesigned following different mechanisms. Therefore, this section does not present the details of a single participatory strategy. However, it is the result of multiple knowledge-generation processes that have occurred collectively to give answers to specific actuations (concrete interventions, workshops, research projects, etc.). Those specific participatory mechanisms have been designed and completed with specific objectives during more than 20 years of research trajectory.

The target audience of DigiHEALTH is improving the health and quality of life of older adults, and on that objective, it has a threefold purpose: (a) minimize barriers related to the use of digital solutions in older adults, (b) enhance the healthy living of a population that lives independently and presents a wide spectrum of characteristics, and (c) facilitate the monitoring and treatment of specific impairments. This digital suite presents a living environment that evolves in a manner that will be improved and enlarged with new solutions that help long-term and healthy living. This will be carried out by continuing the practice presented under the DigiHEALTH theory of change mechanisms (see Section 2.1) that will allow a continuous understanding of stakeholder needs, a continuous adaptation of already presented digital solutions (see Section 3), or even increasing the actual portfolio with other tools that promote health and well-being of older adults.

In this context, we define a three-step methodology for the codesign of digital solutions where multiple stakeholders take an active role with the ultimate objective of improving the quality of life of older people achieving a notable social impact (Figure 1).

### 2.1. Understand Stakeholder Needs

The learning process behind a social intervention, as it is health, is not generally the result of understanding the needs of a specific intervention, but it generally requires a continuous and dynamic approach. As such, the development of a digital suite that is alive and useful requires a continuous learning process. For example, when thinking about desirable functionalities, a reflexive approach that focuses on both a collective learning process as well as on the results of institutional and societal change will be more applicable. Figure 2 provides an example of how a reflexive approach has been applied in a dynamic process that has involved multiple actors, several projects, and many specific interventions. Each cycle of the reflexive approach outlined below contains four steps: observe, analyze, reflect and understand, and report for adaptation.

The digital suite is, therefore, able to integrate traditional performance-based approaches together with a reflexive understanding. Flexible development leads to include tailored solutions that are identified following periodic cycles of reflection to better understand what is working, what is needed, or what is being demanded by healthcare institutions and/or particular users.

The basis for applying reflexive understanding is to facilitate a transparent dialogue that includes transversal and transectorial approaches while keeping the end user in the center. Health and well-being promotion is, indeed, a social intervention by itself; it is a continuous learning process where different stakeholders need to undertake a cyclic reflexive approach. This has been continuously carried out for thousands of years; the innovation led by multiple research groups more recently is about including digital tools in this continuous reflexive approach. The result of this continuous participatory mechanism with respect to the DigiHEALTH suite can be called the DigiHEALTH theory of change as it allows to naturally depict the interactions, causal relationships, internal parameters determining the relationships, and external conditions and assumptions that will draw the future and fate of health digitalization.

### 2.2. Define Main Properties of the Digital Suite for Integration in Wider Digital Ecosystems

The digital suite DigiHEALTH strives to provide health professionals with data and data analytic results that facilitate advanced healthcare services. Indeed, it is not only about improving a set of specific advanced services, but it is also about providing mechanisms to harmonize these results and their impacts and about coordinating the efforts of multiple agents that quite generally face the problem of silos. With this ultimate objective in mind, the DigiHEALTH suite moves forward with the following standardization mechanisms:Adopt standard data models that are aligned with the current necessities of health services. For example, we adopt the Fast Healthcare Interoperability Resources (FHIR) standard that facilitates the exchange of healthcare information in different computing systems regardless of how it is stored. FHIR is a data standard proposed by Health Level 7 (HL7), an organization dedicated to providing a comprehensive framework and related standards for the exchange, integration, sharing, and retrieval of electronic health data (all related information available at http://www.hl7.org/fhir/ (accessed on 1 June 2023)).Some specific solutions (especially those focused on alleviating health problems) are aligned with clinical protocols so that their results will articulate similar reasoning processes as traditionally followed by clinical professionals.Facilitates the usage of SymbIoTe (symbiosis of smart objects across IoT environments) European project for generating smart networked devices, wearables, sensors, and actuators among various IoT domains (all information available at https://www.symbiote-h2020.eu/ (accessed on 15 June 2023)).The digital solutions (DSs) of DigiHEALTH connect with an authentication system based on a single sign-on (SSO) solution for authentication, providing users with a PASETO (Platform-Agnostic Security Tokens) token.The different DSs are prepared to be connected with data lakes or other storage layers by RESTful APIs, which would include data curation and aggregation mechanisms as well as the necessary middleware for its integration with different information services.Specific tests for the integration of DSs at specific systems have been designed. This allows the traceability of the data towards the conclusions and can be readapted to ad hoc applications for specific services.Most of the DSs have been developed using docker technology which facilitates the adaptability and reuse of full or part of the developments. The implementation occurs simultaneously in different parallel environments so that the response to specific requirements from health professionals is fast and scalable at the lowest possible cost.

### 2.3. Codesign of Digital Solutions

Once the main stakeholders’ needs are undertaken and the main characteristics of the DigiHEALTH architecture are defined, the specific characteristics of each digital solution are defined following a quadruple helix (Figure 3) approach [39] that includes stakeholders representing public authorities (regional health services, public hospitals, etc.), health industry (both digitalization entities with the role of technicians, as well as private health providers), academia and specific research groups, as well as citizens that in this case include, healthcare professionals and older individuals.

This participatory paradigm is generally applied in complex projects that involve the transformation of public spaces, governance of sectoral clusters, etc. In this case, it has been applied in the design and development of a DigiHEALTH service that is alive and continually evolving. Indeed, this general understanding is very close to real-life necessities and keeps updated thanks to the following:Participation in clusters and networks helps keep the innovative character of the digital suite and understand the main challenges in terms of health promotion, caregiving, etc., but also in terms of interoperability, modularity, standardization, etc. It is also important to share the results with other stakeholders and keep them close to reality.Development of projects with direct transferability to an industry that provides the context for the codesign of the solutions, the atmosphere to share the experience of testing, and the methods and tools for the validation of the specific features.Collaboration with large consortiums where multiple stakeholders pop up with different interests and necessities. That makes it easier to work with a wide audience of older individuals, test the solutions in different countries that use different languages, and bring forward different cultural assets.Peer to peer meetings with professionals of different fields where caregiving to older people is at the center. Indeed, these close spaces provide the atmosphere to share specific details, the experiences of daily caring, the requirements that would make the tools more friendly or attractive, etc.Design of validation protocols together with health institutions with real-life cohorts. Indeed, health institutions are crucial stakeholders since they provide the methods for testing and validating the solutions before any of them are put in place.Codevelopment spaces where health professionals share common issues of technicians and characteristics of real scenarios. Given certain occasions by innovation projects or in the context of specific actuations or implementations, the communication between health professionals and technicians needs to be fluid.

Older individuals are involved thanks to their participation in several research projects. For example, the personalized nutritional system was designed thanks to a group of 998 older adults (505 men and 493 women); the well-being assessment tool and the recommendation system were codeveloped with 60 people from 4 different countries (Greece, Italy, Spain, and Germany) through 8 use cases of different characteristics. The ultimate objective is to reach a wide audience of older individuals with very different understandings that feel listened to and are helped in their daily activities by the usage of specific digital solutions. The target audience is indeed a wide spectrum of individuals in terms of sensitivities, physical abilities (or impairments), different daily routines, healthy (or unhealthy) habits, etc. The challenge consists of properly negotiating the interests of different stakeholders, but of course, those interests were driven by a common purpose: improving the quality of life of older individuals. That is, different interests are shaped in view of providing a digital suite of solutions that are accessible, friendly, and useful for the promotion of the health and well-being of older adults.

## 3. Results

This section presents the current layout of a digital suite that will be continually updated, improved, and enlarged. At the current stage, the digital suite is available at https://www.vicomtech.org/en/business-solution/solution/healthy-living-and-ageing (accessed on 1 June 2023).

This paper presents a total of eight digital solutions that have been specifically designed and adapted to be used by older adults. Some of them (FaceCOG or ADILIB, for example) could be easily adapted to be used by other public, such as for example, people with specific disabilities. However, this digital suite gathers solutions specifically codesigned for older adults as the main target audience. More specifically, the eight digital solutions respond to three specific objectives that have been identified as a result of the codesign methodology. Indeed, the quadruple helix approach has shown the necessity of (a) minimizing barriers to digitalization, (b) providing solutions for active and healthy aging, and (c) helping health professionals with solutions for specific impairments. The introduction to each section makes explicit the conclusions driven from the participation in multiple forums and clusters as well as the organization of cocreation strategies, design thinking dynamics, etc.

### 3.1. Digital Solutions for Minimizing Barriers of Digitalization in Older Persons

Although generations of older adults are continually evolving and becoming more keep on technologies, research professionals also need to approach their needs and facilitate them to get into new digital scenarios.

#### 3.1.1. Authentication System for Older Adults—Facecog

FACECOG provides an authentication system for older adults that is user-friendly and provides secure face enrolment and verification mechanisms. The solution is based on a face recognition (FR) system that provides additional functionalities to assist users during the facial biometric data extraction process alleviating possible problems related to mobility or reduced interaction capabilities.

FACECOG is a module from Vicomtech’s Viulib library (http://www.viulib.org/ (accessed on 1 June 2023)) designed to support the process of user authentication interacting with IoT devices. It has C++ and Python APIs that allow building apps that would run this functionality locally (i.e., no cloud support required for the calculations) in PCs, mobile phones, tablets, and embedded systems, such as NVIDIA Jetson boards and the Raspberry Pi 4 board. This system facilitates the integration of FACECOG in an Internet of Things (IoT) environment where several devices (e.g., smartphones, tablets, robots, etc.) are expected to interact. It facilitates the following:Compatibility: FACECOG is accessible from various devices. This includes ensuring that the technology is compatible with the hardware and software of other devices, as well as ensuring that those have sufficient processing power and memory to run the face recognition algorithms.Ease of use: FACECOG provides feedback to obtain correct positioning and tips with clear instructions. It also provides the means to interlink with simple and intuitive interfaces.Data privacy and security: FACECOG data privacy and security by guiding end-users on several steps that prevent vulnerable data breaches or identity theft.

Several participatory mechanisms were organized to understand the needs of older adults. Following the quadruple helix approach and accordingly to the requirements of different actions (projects, workshops, etc.), several suggestions for the FACECOG solution were compiled. In the last term, FACECOG has been codeveloped in view of the needs of older adults and considering the most innovative technical resources. The main objective was to facilitate the authentication mechanisms in a diverse IoT environment. The diagram below shows the FACECOG user authentication workflow (Figure 4). It involves three computing components: (1) a remote server, (2) a local processing unit, and (3) an interaction device. In some cases, the local processing unit and the interaction device may be the same device if it has sufficient computing power to run the FR system. Alternatively, the local processing unit could be an IoT gateway that runs the FR system and communicates with the interaction device by exchanging messages (e.g., images and responses).

The full user authentication process relies on a token-based authentication system [41], which is supported by FACECOG to make it easier for older adults to use the system, provided certain conditions are met. The user must first register on the platform using a user ID and password and then complete the facial registration process. To assist with this, the system detects and analyzes facial cues in front of the camera to provide guidance on how to position the face for a suitable image to be captured [42]. During the login process, if a token is available, FACECOG follows a similar procedure to the facial data registration process to verify whether the detected potential user’s face matches the expected person. While it is not possible to design a completely secure system against all kinds of threats, it is possible to create an appropriate level of security for a specific scenario and expected types of attacks. In our approach, biometric data is never transferred to the cloud and is only stored on devices where the FR solution has been deployed. Therefore, a potential attack could be someone accessing the devices, robots, or gateways to steal the stored data. To prevent this, the data should be encrypted, and the encryption keys should be kept safe by an administrator. In our context, we use fully homomorphic encryption [43] to enable the matching to be performed directly in the encrypted domain. By using this method, biometric data shared between devices, robots, and the gateway remains encrypted as long as the administrator keeps the encryption keys secure (e.g., in a secure hardware element [44]).

#### 3.1.2. Digital Voice Assistant—Adilib

ADILIB is a built-in software for creating and developing digital voice assistants [45,46]. An ADILIB conversational agent is capable of understanding natural spoken language and generating a response. Through its dialogue management system, previous interactions are stored and may influence the conversation by helping make adequate context-aware dialogue decisions.

ADILIB has been tailored for older individuals who require a certain degree of support and care from professionals but wish to stay in their own homes rather than in a care facility. The motivation behind the digital suite is to facilitate personalized monitoring by caregivers of older adults living at home through a spoken interface they find easy to use. In this sense, ADILIB provides two functionalities that were specifically designed to enhance the experience of both older adults living at home and their caregivers. First, we introduced a wake-up word [47], a mechanism that starts the conversational agent only when a predefined word or expression is uttered. The user can therefore be reassured knowing that the assistant only listens (and gathers data) once it has been activated via the wake-up word. The second is the introduction of skills and conversation structures that can be adapted for each conversational agent and customized for each patient. Four different types of skills have been implemented:Agenda: The agenda allows managing the patient’s schedule to keep track of all kinds of events, such as medicine intake or doctor appointments. Caregivers can create and manage calendar events that can then be consulted by the patient at any time (for instance, by saying, “I want to check my medical appointments for today”).How-to: the how-to skill guides the patient through a series of—often sequential—processes in order to help them accomplish a task. The caregiver can easily program new tutorials (and even import tutorials from the WikiHow project) that are suitable for the patient, such as “how to make a WhatsApp call” or “how to relax”.Questionnaire: This skill allows the caregiver to create and manage questionnaires to be completed by the patients in a conversational way. Caregivers can use the information gathered from customized questionnaires to monitor the patient’s progress and adjust their treatment if needed.Reminder: Finally, reminders are a special type of skill that interacts with the agenda, how-to, and questionnaire skills, activating them at a specific date and time defined by the caregiver. Through this skill, patients can be reminded of the events in their calendar, such as medicine intake or medical appointments, or be prompted to perform certain tasks, such as completing daily follow-up questionnaires.

Caregivers in charge of older adults can employ the developed skills to personalize the digital voice assistant with agenda appointments, how-to tutorials, and questionnaires, etc. Caregivers can also use the reminder skill to program reminders and make sure about attending important agenda events and/or answering follow-up questionnaires.

While creating patient-specific skills may be time-consuming for caregivers at the beginning, our experience in various demonstration sites has shown their positive acceptance and impact.

### 3.2. Digital Solutions for Active and Healthy Aging

The holistic understanding of health implies being aware of health and well-being and actively participating in equilibrating our routines and daily life. This section presents two digital solutions that help continuous monitoring and self-calibration of health metrics.

#### 3.2.1. Well-being Assessment

This digital solution allows the detection of well-being disturbances in the daily activities of individuals through a personalized threshold based on users’ routines [48]. The objective of this method is to aid professionals and caregivers to early detect abnormal conditions that may be related to specific frail episodes or deterioration.

The system incorporates different functionalities:Giving risk values to each of the individuals to detect health conditions and priorities.Creating alerts for low- and high-risk alterations, detecting irregularities, and limiting or preventing their worsening.Comparing a detected alteration with other individual’s attributes in time.Analyzing and preventing the development of any complications.Dashboard for visual analytics that incorporates filters and other features.Well-being questionnaires for validating the detected alterations.

To develop those functionalities, it is necessary to collect measurements that provide information about the lifestyle and well-being of older individuals over time. To this end, information regarding older people’s activity, sleep quality, fluid intake, and vital signs was captured by different methods, such as wearable devices, questionnaires, or even through a digital assistant specifically designed with this aim (see Section 3.2.1). The technical architecture of the well-being assessment system is shown in Figure 5. It has been designed to process the information by following three sequential steps:Preprocessing the data making sure that metrics are relevant in relation to well-being assessment.Compare daily metrics with data records of previous days and compute z-score.Identify abnormal and critical conditions as those that may represent some health-specific conditions or derive from problems.

The estimation of the abnormal and critical conditions is the main output of the well-being assessment tool. Those are estimated as anomalies in the regular health condition by adopting z-scores on the different health metrics (physical exercise, for example). The results are then validated by using specifically designed questionnaires so that end-users are directly asked to corroborate the assessment output. In this manner, (i) the quality of the well-being assessment condition can be ensured, (ii) the reasons why the alterations occur can be gathered and tracked, and (iii) some feedback can be used by researchers in order to calibrate or improve the overall assessment method. This validation step is especially relevant in health applications where artificial intelligent engines are working on specific (and personal) health metrics. The questionnaires are codesigned by multiple stakeholders, caregivers, end-users, technological partners, etc., so that we ensure that they are suitable for every specific use case.

The final overview of all this information can help researchers and professionals to assess the general well-being, identify possible deterioration, or see whether some tendencies are arising. The visual analytics are displayed in a dashboard (Figure 6) that represent the main well-being assessment tool. In this case, the application shows the metrics about sleep, physical activity, and liquid intake. Although social contact has been shown to have an important effect on well-being, this particular use case does not include it.

The well-being assessment tool, together with other digital solutions available through the DigiHEALTH suite, provides a great context to talk and interchange views and perceptions about the experience of well-being with caregivers, technicians, etc. Older individuals are empowered by using advanced digital solutions; they can now keep track of several metrics and have the possibility to discuss about their health and well-being using specific metrics or health data.

#### 3.2.2. Recommendation System for More Efficient Healthcare

The recommendation system provides personalized well-being recommendations generated for each individual. The goal of these recommendations is to improve the quality of life and promote healthier habits among the older population. As a result of this, a better health status and a reduction in healthcare services can be achieved.

The main component of the recommendation system is a rule engine that is responsible for the generation of personalized messages based on the evaluation of daily activity or well-being metrics. The recommendations are formalized by using a logic system (rules) that may be adapted for each case study. To facilitate the scalability and adaptability of the solution, the solution is built upon an authoring tool that allows for the creation, adaptation, and further management of the formalized knowledge [49]. The use of a rule engine ensures the scalability of the solution, assuring a fast evaluation of data independently of the complexity of the logic behind the result or the number of variables to be evaluated.

The exchanges of data follow the FHIR standard for clinical data exchanges, which guarantees the interoperability of the system. Incoming and outgoing data is shared via HTTP requests, which are mediated by the API of the recommendation system. The system can receive data from different data sources, which in turn allows it to integrate different typologies of data and generate recommendations for different domains. As a result of this, the intervention domains of the recommendations can be expanded towards a more holistic- and person-based approach. In the current version, the data used includes the daily intake of several beverages and activity data such as the daily toll of steps and records about the sleep duration (Figure 7).

The entire data flow generated by the recommendation system can be checked and analyzed (Figure 8). This allows researchers or clinicians to inspect the list of recommendations and take further steps in communicating the results, evaluating the impact, designing healthcare services, etc.

#### 3.2.3. Personalized Nutritional System for Older Adults

This digital solution presents a system designed for health professionals with the objective of aiding them in the prevention of malnutrition in hospitalized older adults. It is composed of two different tools, (i) a nutritional recommender system for the prevention of nutritional issues and (ii) an AI-based model for the prediction of malnutrition risk in older adults.

##### Nutritional Recommender System

This functionality provides health professionals and caregivers with a nutritional decision support system that considers not only the different nutritional needs (e.g., malnutrition risk, intake level, or the need for texture-adapted meals) but also the whole environment of an older patient, such as sociodemographic factors (e.g., sex, age…), psychosocial factors (e.g., psychosocial disorders), and morbidity factors (diseases). In addition, the personalized nutritional guidance [50] that offers this solution is divided into two types of recommendations, i) dietary and nutrition guidelines and ii) menu suggestions. In the first type, diet, fortification, food and liquid adaptation, supplementation, enteral nutrition, and follow-up areas are managed, whereas in the second type, different examples or suggestions of menu types and their ingredients are offered. However, the domain-independency and the capacity to use standard formalization protocols that offer our system, together with the authoring tool [48], make these guidelines extensible and adaptable to different use cases (Figure 9).

##### Malnutrition Risk Predictive Model

After identifying the key factors that contribute to malnutrition in older adults in a previous study carried out by our research group [50], those variables were used to develop an AI-based model for predicting the risk of malnutrition for this population. A total number of fifteen variables was introduced in the prediction system, and a three-optional response was obtained: low, medium, and high risk of malnutrition. In order to do that, an easy-to-use interface was made available for professionals (Figure 10).

### 3.3. Digital Solutions for Specific Impairments

#### 3.3.1. Heart Failure Decompensation Predictive Model

Heart failure (HF) is a clinical condition caused by a structural and/or functional cardiac malfunction. Patients with HF suffer decompensation, a clinical condition in which a structural or functional alteration of the heart results in a lack of capacity to transport (or eject) blood within the required physiological pressure levels. This leads to functional limitations requiring immediate therapeutic intervention [51].

HFPred is a digital solution that detects or predicts decompensations that often occur in individuals that have suffered heart failure (HF) episodes. Since early detection causes further worsening of health conditions or even death, this solution tries to anticipate and foresee the most common decompensation issues. More specifically, the computational engine estimates the risk of patients with heart failure to decompensate in the next seven days, given daily vitals and an eight-question questionnaire. The artificial intelligence engine has been trained with 242 patients that participated in specific time frameworks from 2014 to 2018. The training and results of this model are explained by Larburu et al. [52].

Figure 11 shows the dashboard with the list of patients with the estimated risks. The solution ranks the patients at the highest risk for a given day, providing the opportunity to communicate with the patients or articulate more specific measures.

For a more specific analysis of a given patient, the HFPRED tool provides clinicians with all available information, including time series data of specific health parameters or last update responses to the questionnaire (Figure 12).

#### 3.3.2. Gait Analysis for Motion Quality Assessment in Older Adults

According to the World Health Organization, 28–35% of people over 65 years fall each year. One of the biological factors associated with an increased risk of falls is the decline of physical condition that can be reflected objectively in terms of gait performance loss [19,20]. Monitoring the gait of older adults can provide insight to caregivers on whether a rehabilitation program is effectively improving the participants’ gait performance.

A cost-effective and widely used device to obtain gait-related biomarkers is the inertial measurement unit (IMU), a wearable sensor that measures triaxial acceleration and angular rate with an accelerometer and a gyroscope, respectively. To integrate a gait assessment tool into the clinical environment, the sensor setup must be minimalist and easy to use by nontechnical persons, and the algorithms must be validated with a representative population (e.g., older adults with orthopedic or neurological pathologies using different walking aids).

The gait analysis tool [19] adapts several algorithms to extract a set of gait spatiotemporal parameters with only one IMU located above the ankle. The main metrics of interest are stride time, swing-stance ratio, stride length, and stride velocity. If data from two IMUs are available, one at each ankle, the tool will also provide asymmetry metrics. The gait assessment tool has been developed with two datasets collected at a nursing home. The 28 participants are aged over 80 and have different degrees of mobility: walker-aided, with parallel bars, crutches, a cane, and without any external aid. Part of the cohort suffers from different neurological pathologies (stroke in the previous years, normal pressure hydrocephalus, dementia, etc.) and orthopedic disorders (gonarthrosis, lumbar spinal stenosis, hallux valgus, etc.). A mean gait pattern is extracted from the angular velocity data of each participant, which can visually help the physical therapist in assessing the gait.

Figure 13 shows examples of a walking pattern of an older adult walking without aid (left) and an individual walking with parallel bars due to normal pressure hydrocephalus (right), a neurological condition that is characterized by gait apraxia, thus the inability to lift the feet. We have computed a quantitative metric to assess the curved shape of the gait pattern between the main events of a stride, the heel strike (first local minima), and the toe off (second local minima). This metric is called the shape index and is calculated from the minimum and maximum curvatures of the angular velocity data. The metric is normalized in the range 0–1, meaning that a parabola-shaped curve will have a shape index of one (Figure 13a), and a relatively flat-shaped curve will have a value close to zero (Figure 13b). Figure 14 shows a boxplot of the shape index values for one of the datasets available.

The boxplots in Figure 15 show a comparison of some spatial (upper row) and temporal (lower row) gait metrics when stratifying subjects by walking aid. We can see that the parallel bars group has a slower gait (see stride velocity on the upper right) with considerably shorter steps (see stride length on the upper left) than the other two groups. The swing ratio metric on the bottom right represents the percentage of the swing phase in the total stride. Normally, this value is around 40%, close to the mean value for the group without aid, whereas for the parallel bars group is only 33%; hence the stance phase is longer in this group.

#### 3.3.3. Face Gesture Recognition System for Patients with Degenerative Diseases

Orofacial Rehabilitation is a specialized branch of physiatry that aims to address physical impairments in the orofacial system [53,54]. This system comprises the organs that are responsible for various physiological functions such as breathing, swallowing, speaking, phonation, and facial expressions. Facial gesture recognition systems can be beneficial for older adults in various ways. These systems can help older adults improve their facial muscle control and facial expression, which can be particularly important for those who have lost muscle control due to aging or neurological conditions such as stroke. In addition, facial gesture recognition systems can provide older users with a convenient and engaging way to practice their facial exercises. By using a system that gives feedback on their progress and encourages them to continue, older users may be more motivated to stick with their rehabilitation program. Lastly, facial gesture recognition systems may also be able to assist older users in communicating more effectively. For instance, if a person is unable to speak due to a stroke or other condition, a facial gesture recognition system could help them communicate through facial expressions or gestures. This could be particularly helpful for individuals who are isolated or have limited social interaction.

For this reason, we developed the OROFACE approach to address the unique needs of orofacial rehabilitation [54]. One of the main challenges in designing such a system is that traditional facial gesture recognition methods often do not cover the wide range of facial gestures that are commonly used in orofacial rehabilitation sessions. As a result, it can be difficult to create a system that is able to accurately recognize and interpret these gestures. Thus, OROFACE aims to address the following tasks:Construct a balanced facial image dataset that includes various facial appearances and features a range of facial gestures commonly used in orofacial rehabilitation, including neutral expressions. This dataset will serve as the reference for measuring the trainee’s progress.Develop an effective and efficient facial image processing strategy for frame normalization and motion magnification, particularly for micro gestures.Design personalized metrics for evaluating facial gesture accomplishment, taking into account each individual’s neutral expression.Analyze the visual distinguishability of the required facial gestures in order to train an effective and efficient facial expression classification model.

The first task involves recording a professional expert performing a training session for orofacial rehabilitation, as well as other people replicating the session while watching it. These videos are then segmented to extract sequences of the full facial gesture action and separate sequences of the neutral expression. The second task involves using contrast-enhanced normalized differential images (CENDIs) to enhance relevant areas of the gestures [53]. The third and fourth tasks involve using deep neural networks (DNNs) to classify the gestures based on CENDIs generated from the previously mentioned dataset. However, in some cases, it may be necessary to fuse similar gestures and retrain and retest the DNN until an effective classifier is obtained. The goal is to measure the degree of achievement of a canonical reference, and lightweight DNNs with a good balance between accuracy and computational complexity will be used for this classification and detection of facial regions and landmarks on devices with limited resources.

Figure 16 shows examples of different orofacial gesture performances analyzed by OROFACE. The bar size indicates the level of degree of achievement obtained by the user. In these examples, the first three gestures are correctly performed, while the last one does not fully match any of the predefined gestures.

## 4. Conclusions

This paper presents a suite of digital solutions for long-term healthy and active aging. Digihealth (https://www.vicomtech.org/en/business-solution/solution/healthy-living-and-ageing (accessed on)) is the result of an extensive trajectory of research for the digitalization of the caregiving sector, which is particularly sensitive and complicated. Working in a quadruple helix approach facilitates a reflexive approach where public authorities, industry, academia, and older adults are called to participate. The result is presented in the form of eight digital solutions that carry out the following:-Minimize digital barriers for older adults: (1) authentication system based on face recognition and (2) digital voice assistant.-Facilitate active and healthy living: (3) well-being assessment module, (4) recommendation system, and (5) personalized nutritional system.-Mitigate specific impairments: (6) heart failure decompensation, (7) mobility assessment and correction, and (8) orofacial gesture trainer.

These codesign solutions move forward standardization mechanisms by adopting the FHIR standard, facilitating the usage of SymbIoTe (symbiosis of smart objects across IoT environments) European proposal, connecting with an authentication system based on a single sign-on (SSO) solution, etc. Specific details about specific libraries are available at DigiHEALTH, a live website that will be continually updated and enriched with further experiences of collaboration.

It seems clear that digital health is being demanded as a promising way to solve the challenges posed by rapid global aging. This paper pays special attention to the technical part and the codesign process of digital health. Future studies will pay more attention to the social, legislative, and ethical perspectives of digital health.

## Figures and Tables

**Figure 1 ijerph-20-06200-f001:**
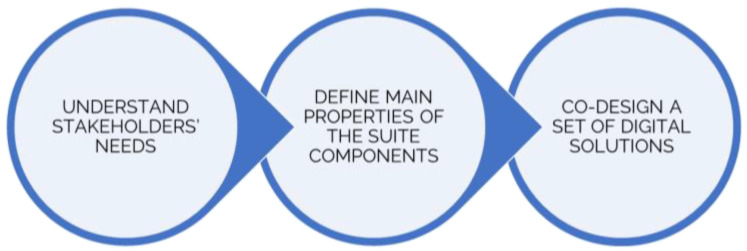
Three-step methodology for the continuous improvement and update of digital suite of solutions for healthy and active aging.

**Figure 2 ijerph-20-06200-f002:**
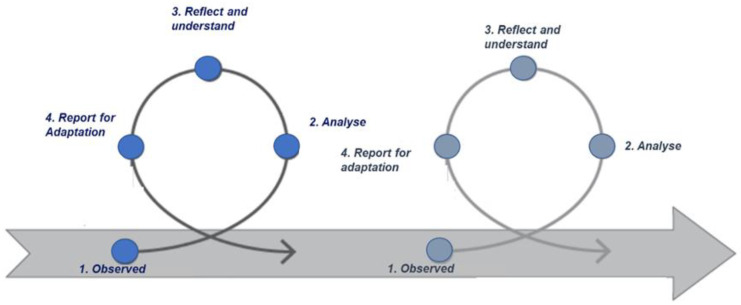
Four steps of the reflexive approach cycle.

**Figure 3 ijerph-20-06200-f003:**
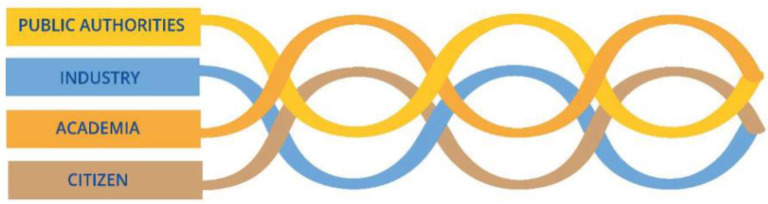
Quadruple helix approach [39]. Figure from quadruple helix model [40].

**Figure 4 ijerph-20-06200-f004:**
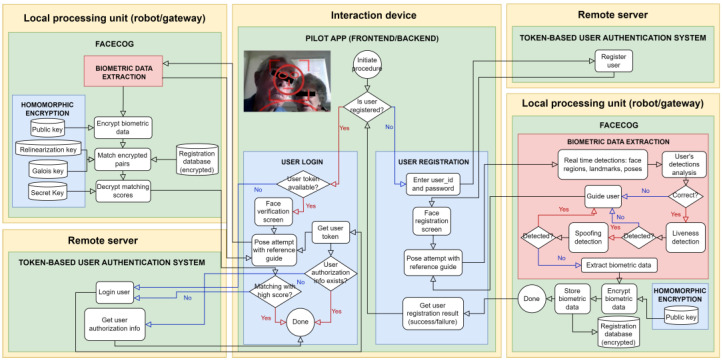
FACECOG’s user authentication workflow diagram.

**Figure 5 ijerph-20-06200-f005:**
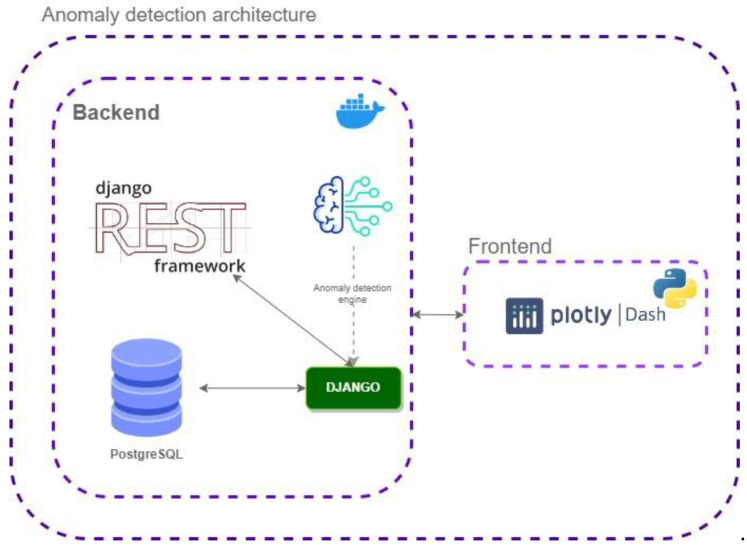
Technical architecture of the well-being assessment system.

**Figure 6 ijerph-20-06200-f006:**
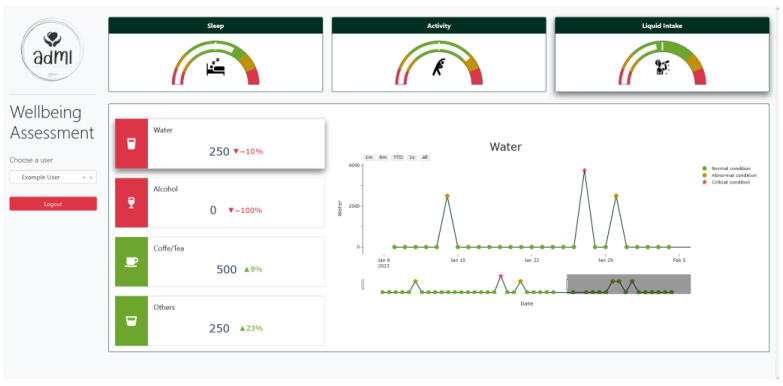
Interface example of the well-being assessment digital solution.

**Figure 7 ijerph-20-06200-f007:**
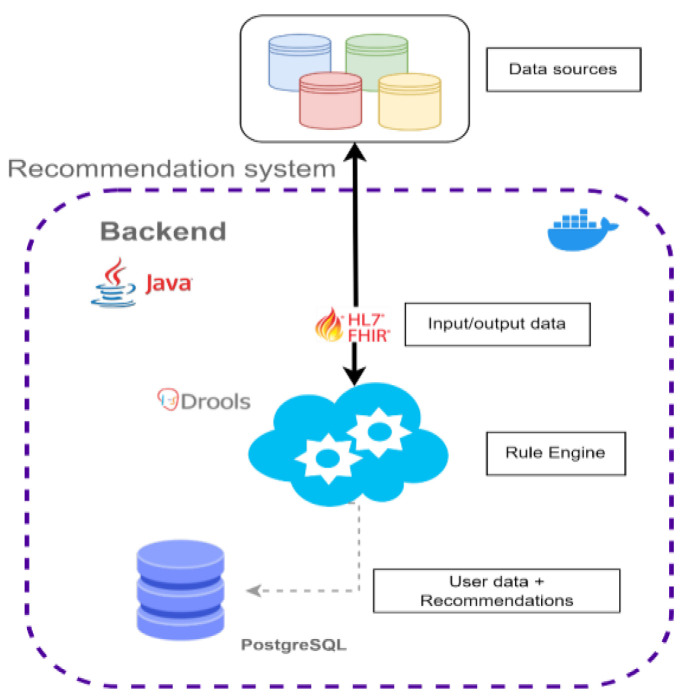
Architecture overview of the recommendation system.

**Figure 8 ijerph-20-06200-f008:**
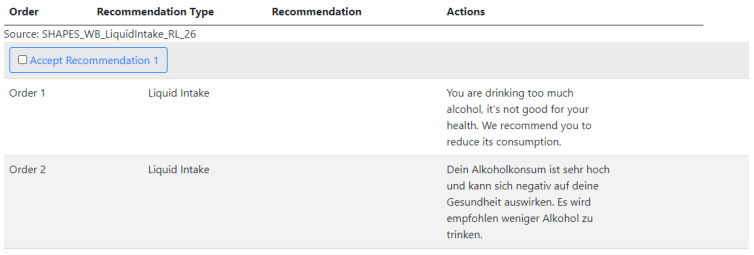
Example of recommendation generated for end users with respect to liquid intake.

**Figure 9 ijerph-20-06200-f009:**
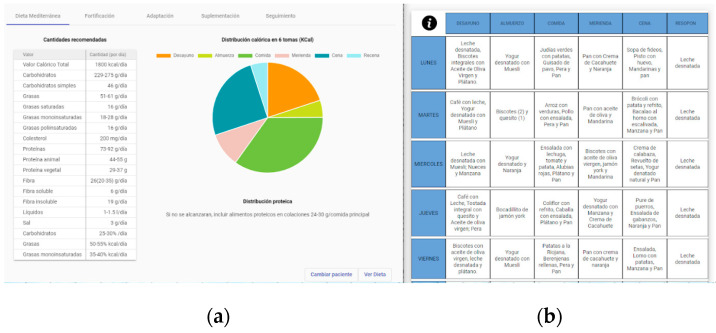
Example of the visualization of nutritional guidelines (**a**) and menu suggestions (**b**) in a web application for health professionals.

**Figure 10 ijerph-20-06200-f010:**
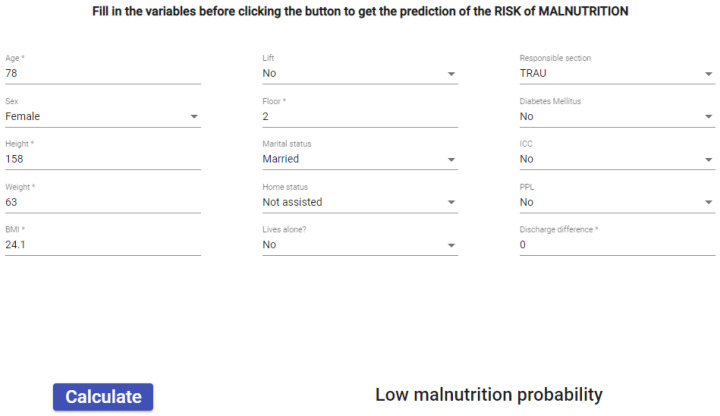
Interface for the evaluation of the risk of suffering malnutrition.

**Figure 11 ijerph-20-06200-f011:**
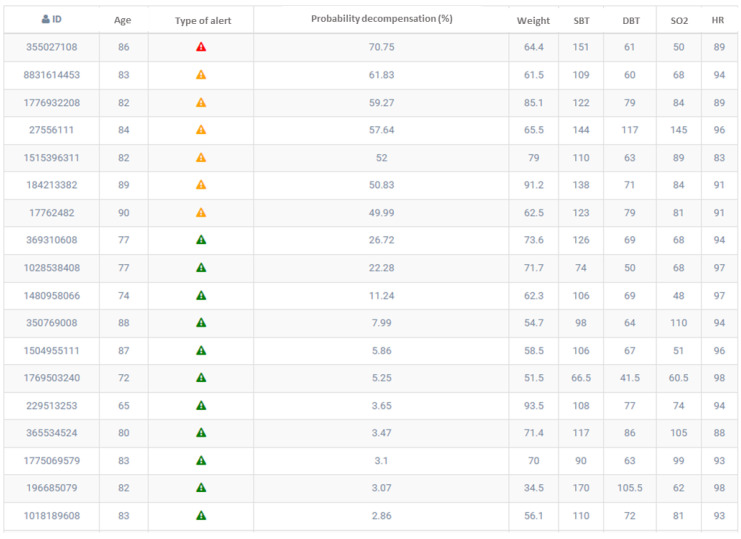
List of patients with their corresponding risk of suffering decompensation.

**Figure 12 ijerph-20-06200-f012:**
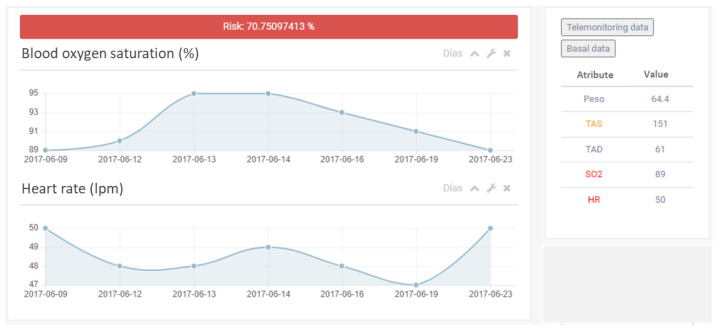
Dashboard with detailed information about a given patient, including the estimated risk and most significant variables.

**Figure 13 ijerph-20-06200-f013:**
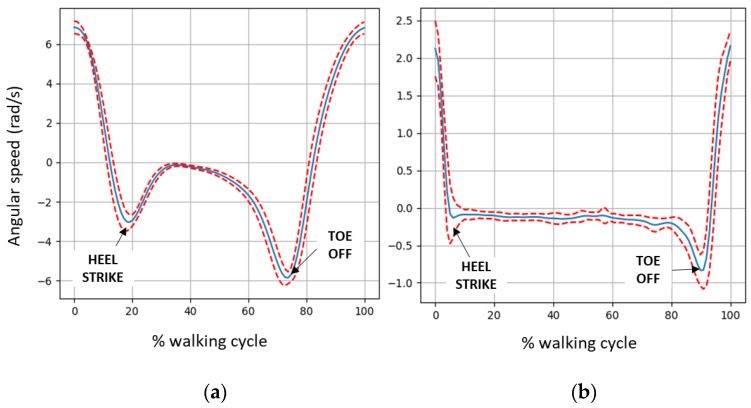
Mean (solid blue) and standard deviation (dotted red) of angular speed in walking direction for an individual without walking aid (**a**) and using parallel bars (**b**).

**Figure 14 ijerph-20-06200-f014:**
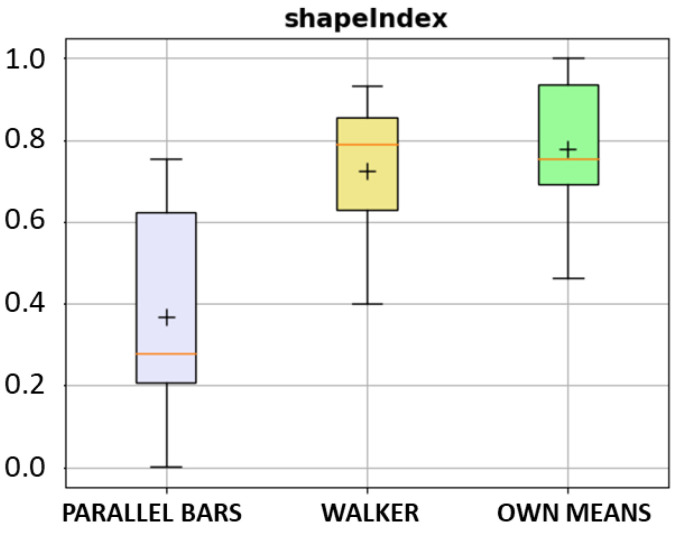
Boxplot for shape index metric stratified by walking-aid, reflecting the different gait-pattern curves.

**Figure 15 ijerph-20-06200-f015:**
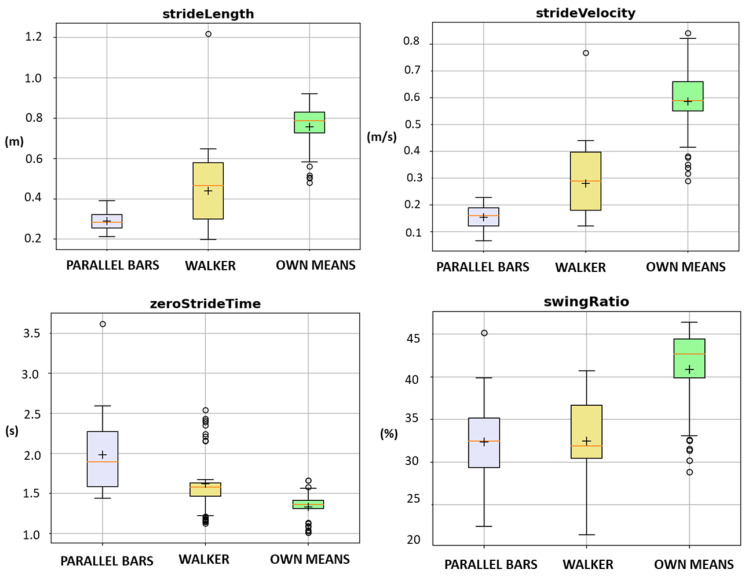
Boxplots for spatiotemporal metrics stratified by walking-aid for one of the datasets available: parallel bars (n = 26 strides), walker (n = 63 strides), own means (n = 50 strides).

**Figure 16 ijerph-20-06200-f016:**
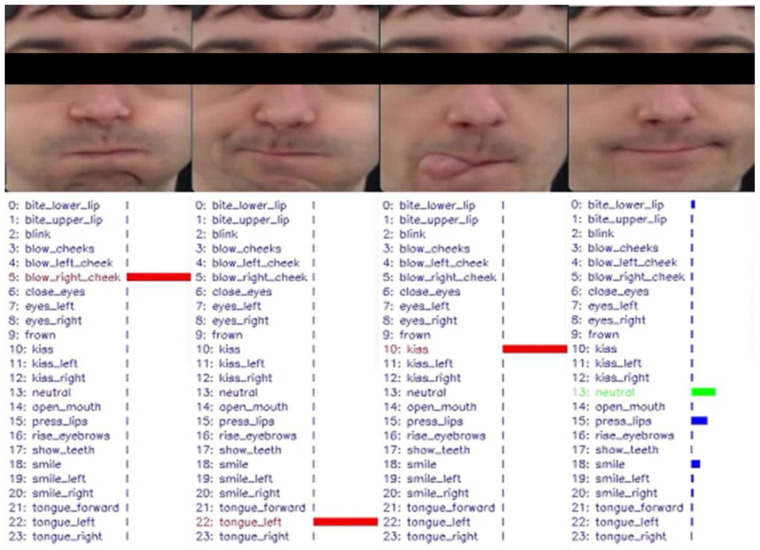
Examples of orofacial gesture performance evaluation with OROFACE.

## Data Availability

Further details and data will be found at https://www.vicomtech.org/en/business-solution/solution/healthy-living-and-ageing.

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
