# Peer review of "DigiHEALTH: Suite of Digital Solutions for Long-Term Healthy and Active Aging"

_ijerph, 2023, doi:10.3390/ijerph20136200_

Round 1
Reviewer 1 Report
The authors describe a digital suite which combines eight different applications which can be used to improve caregiving and/or older people’s health and wellbeing. The paper includes an introduction, a methods and materials section, the description of the different applications and a short conclusion. It brings together different types of applications that focus on different types of (older) patient/user groups. While it for sure is helpful to combine several applications, the process of developing these (participation) is unclear, it is not clear how knowledge on the needs of older users was obtained and how conflicts between stakeholders were managed. The article is presented in an introduction - methods - results -conclusion format, but the results present the actual applications rather than the process described in the methods section. It is not a research article but a presentation of digital tools that can be combined in the suite. I have listed other points in the text where I felt that information was missing.
Page 2: ‘Improving the quality of life implies extending the period of time in which one maintains him or herself an active member of the community, contributing to society and enjoying life.’
I believe that this statement needs some further explanations. What do you consider being an active member of the community, how do you define contributing to society and how is this linked to enjoying life? It is important not to overemphasize ‘activity’ and ‘contribution’, since different people are not similarly able to achieve both using narrow definitions.
Page 2:’Ensuring federal spaces for data sharing will facilitate a democratic use of information where data ownership is ensured at the same time that specific professionals can make use of the information and provide advanced or improved services.’ Please expand on how federal spaces facilitate democratic use. Please also provide references. The reference mentioned here does not seem to relate to your argument/topic.
Pave 2: please explain in more detail what you mean by ‘satisfaction of healthcare providers’. Please also provide a reference for the statement starting ‘All of these digital tools will optimize healthcare resources by facilitating an improved experience…..’,
p.3pp. ‘this digital suite presents a living environment that evolves in a manner that it will be improved and enlarged with new solutions that help the long term and healthy living’ (please explain ho this will be done.
p.3 notorious should probably be notable (?)
p.3 Figure 1: who are these stakeholders? How are conflicts among their needs resolved?
p.3 ‘the learning process behind a social intervention, as it is health’ – please explain what you mean by health itself being a social intervention
p.3: Who are the healthcare institutions and/or particular users targeted?
p.4 how are specific solutions aligned with clinical protocols?
p.4 who are the main stakeholders?
p.5: please avoid abbreviations such as etc.
p.5 please explain ‘participation in clusters and networks’
p.5. how are diverging interests of stakeholders negotiated?
p.5 how are older people involved?
p.5 please define what you mean by ‘a wide audience of people’
p.5f. facehog and ADILIB seem to address a disability – why is the suite not aimed at improving the care of people living with a disability
p.6 ‘developed with the needs of older adults in mind’ is not participation; please explain how information about this potential group of beneficiaries was collected
p.7 it seems that ADILIB is developed for living ‘without care’ for someone who can only communicate with their voice; please specify the imagined use, in which situation and with which care would someone find this useful. If someone is not able to communicate with anything other than their voice they will most likely require care. The caregiver could talk with them about their plan for the day. How does this improve their situation?
p.8 is it risk or wellbeing that is to be analysed?
p.7 by whom are the self-designed questionnaires designed?
p.9 wellbeing depends significantly on social contact. How is this addressed by the application or the imagined use?
p.11 figure 10: please translate this into English
Reviewer 2 Report
This paper presents a suite of digital solutions for long-term healthy and active ageing. The article details the features and functionalities of this technological system, but there are some areas that need improvement:
There are multiple instances of non-standard abbreviations and markings throughout the text, such as "FHIR standard""HL7" and "DSs". The author should provide explanatory annotations to clarify these abbreviations. The abbreviation "HF" used in line 422-428 is confusing.
There are missing punctuation marks in several places throughout the text, such as in Lines 141, 144, 147, 154, and so on. The authors should review the punctuation usage in the article.
Several images in the article are blurry and unclear, such as the text of the process diagram in Figure 4, the text in Figure 10, and Figures 13, 14, and 15 affect the readability and understanding of the content.
Reviewer 3 Report
This is a very good paper and addresses a very important issue in elderly care. And digital health is a promising way to solve the challenges posed by global rapid aging. This paper pays more attention on the technical part and the co-design process of digital health. Future studies could pay more attention on the social, ethical, law, etc part of digital health.
Reviewer 4 Report
The paper presents a suite for digital health solutions covering different parts with details. Some sections of the paper need to be improved.
The paper does not cover state-of-the-art methods related to the proposed one, neither in the introduction section nor there is a dedicated section to cover the related works. Including such a discussion will lead to the problem statement and the need to develop the new systems.
The contribution of the paper in terms of the novel aspects i.e., novel methods, techniques, and datasets, generated needs to be clearly stated.
How the validity of the datasets and the results is measured. Are there any baseline results available? Are the results validated by the experts? The details need to be included. Can the results be compared with other methods?
Round 2
Reviewer 4 Report
The revised version covers most of the comments.